# The Financial Toxicity Experience of Patients with Colorectal Cancer During Chemotherapy: A Qualitative Study

**DOI:** 10.3390/curroncol32010023

**Published:** 2024-12-31

**Authors:** Yanli Yao, Shijing Zhang, Qun Yu, Xia Zhao, Xinqiong Zhang

**Affiliations:** 1School of Nursing, Anhui Medical University, Hefei 230032, China; yyl627611239@163.com (Y.Y.); 18895339806@163.com (S.Z.); qunyu47@gmail.com (Q.Y.); m19840176690@163.com (X.Z.); 2Department of Gastroenterology, The First Affiliated Hospital of Anhui Medical University, Hefei 230022, China

**Keywords:** colorectal cancer, chemotherapy, financial toxicity, experience, qualitative study

## Abstract

Background: This study aimed to explore the experience of financial toxicity in patients with colorectal cancer during chemotherapy and to inform the development of targeted interventions. Methods: A descriptive qualitative research method was used to conduct semi-structured interviews with a purposive sample of 15 patients with colorectal cancer undergoing chemotherapy who attended the Department of Medical Oncology of the First Affiliated Hospital of Anhui Medical University from March to June 2024, and the data were organized and analyzed using the Nvivo 11.0 qualitative data analysis software and the thematic analysis method. Results: Four themes were extracted: patients with chemotherapy-stage colorectal cancer bear direct and indirect multifaceted economic pressures, are affected by multidimensional risk factors, which cause multiple adverse outcomes, and cope with financial toxicity in various ways. Conclusions: The experience of financial toxicity in colorectal cancer patients undergoing chemotherapy is presented in a multidimensional format, with multiple causes influencing their financial toxicity. In the future, healthcare professionals should identify patients at high risk for financial toxicity, provide financial toxicity interventions and support, and mitigate their exposure to financial toxicity.

## 1. Introduction

Colorectal cancer (CRC) is one of the most common malignant tumors of the digestive tract. It is estimated that in 2022, the global incidence of colorectal cancer will account for about 9.6% of all malignant tumors and rank third [1]. In China, the number of colorectal cancer cases in 2022 will be 517,000, and the incidence rate will be the second highest among the total number of new cancer cases in China, accounting for 10.7% of the total number of new cancer cases [2]. In recent years, with the advancement of treatment technology and the use of new anticancer drugs, the survival rate of colorectal cancer has been dramatically improved, while at the same time, the cost of prolonged survival has also been increasing [3]. Clinical evidence shows that most of the patients have metastasis when the tumor is found, and surgery or radiotherapy can only provide local treatment, which needs to be combined with different degrees of chemotherapy [4].The concept of “financial toxicity (FT)” was proposed by American scholar Zafar and Abernethy in 2013, which refers to “the objective economic expenditures associated with cancer treatment and the resulting psychosocial distress, behavioral changes, and decreased quality of life, etc. [5]”. Some studies have shown that chemotherapy worsens the financial toxicity of colorectal cancer patients [6]. In addition, some patients receiving chemotherapy are in advanced stages or have severe comorbidities, which leads to higher reported financial toxicity [7].

FT negatively affects the mental health of colorectal cancer patients and their treatment adherence, which in turn interferes with the subsequent treatment process and outcome [8]. International research on financial toxicity has progressed, while domestic relevant research is still in the initial stage [9]. The study of the personal experiences of financial toxicity among colorectal cancer patients undergoing chemotherapy has not yet garnered widespread attention in China. Although financial toxicity is more evident among cancer survivors, financial status is not routinely assessed in clinical practice [10], and there are few evidence-based interventions to reduce this burden. Therefore, this study selected a theoretical model of financial burden after cancer diagnosis [11]. This model includes the causes, moderators, and types of financial burden. It also describes clinical outcomes that may be affected by financial issues. The components of financial burden are physical, psychological, healthcare-specific (provision of treatment), and general (provision of necessities). The model focuses on explaining how financial toxicity occurs. Additionally, it provides a framework for exploring the experience of financial toxicity in patients with colorectal cancer undergoing chemotherapy. This study aimed to conduct an exploratory investigation through descriptive qualitative analysis methods to provide a comprehensive and in-depth understanding of the intrinsic experiences of financial toxicity during chemotherapy for colorectal cancer patients, in order to understand intervention measures for patients’ economic risks.

## 2. Materials and Methods

### 2.1. Participants

The purposive sampling method was used to select colorectal cancer patients who underwent chemotherapy in the Department of Medical Oncology of the First Affiliated Hospital of Anhui Medical University from March to June 2024 as the interview subjects. (1) Inclusion criteria: ① Colorectal cancer patients who met the diagnostic criteria of “Chinese Colorectal Cancer Diagnostic and Treatment Criteria (2023 Edition)” [12]; ② age ≥ 18 years old; ③ knowledge of the disease and diagnosis; ④ people with chemotherapy duration ≥ 3 months; ⑤ good communication skills and voluntary participation in the interview. (2) Exclusion criteria: patients with other serious diseases or psychological or mental disorders in combination were excluded. The survey sample size followed the principle of data saturation. The data tended to be saturated when the 13th patient was interviewed, and 2 additional cases were interviewed, so the sample size was 15 cases.

### 2.2. Design

We adopted a descriptive qualitative research approach grounded in the theoretical framework of economic burden subsequent to a cancer diagnosis [11]. We reviewed the relevant literature and initially formulated the interview outline. We pre-interviewed two colorectal cancer patients in the chemotherapy stage, made appropriate adjustments according to the interviewees’ comprehension of the questions, and finally determined the outline of the interview (Table 1).

### 2.3. Setting and Data Collection

Semi-structured interviews were conducted by a researcher trained in the qualitative research system and with experience conducting interviews. The purpose and significance of the study were introduced to the patients before the interviews, and the interviews were started after the interviewees’ consent. The contents of the interviews were audio-recorded, and all the interviews were conducted in a quiet environment with no external influences. The interviews were conducted in the conference room of the ward. The interviews were conducted using the interview outline, but the order of the outline was not fixed, and the theme of the interview was the main focus. In the interview, the interviewer repeated, reorganized, and summarized the respondent’s answers. They used follow-up questions and rhetorical questions to obtain complete, comprehensive, and accurate information. The interviewer also recorded non-verbal cues like the respondent’s expressions and eye movements. The interview lasted for 30 to 65 min.

### 2.4. Data Analysis

The audio recordings were transcribed into text within 24 h after the interviews. To protect patients’ privacy, the interviews’ results were anonymized, and a numerical number replaced the interviewees’ name. With the use of a three-level coding approach in thematic analysis [13] (i.e., initial coding, sub-theme coding, and theme retrieval), the organized data were coded with the aid of Nvivo 11.0 qualitative analysis software for word frequency statistics and coding organization. In the initial coding process, two researchers independently coded meaningful units, resulting in a complete list. They then merged overlapping content from their respective initial codes into higher-level categories. Next, during the theme coding process, relationships between categories were organized and interpreted, summarizing them into more inclusive categories, known as sub-themes. Finally, by further focusing on different sub-themes, broader themes with greater significance were generated, and all relevant coded data within the determined thematic scope were organized, clarifying the hierarchical relationships among the codes. Throughout the entire process, sharing, reviewing, comparing, discussing, and refining were repeatedly carried out until the researchers reached a consensus.

### 2.5. Quality Control

The entire process adopted peer reporting to verify the reliability and rigor of thematic analysis. Two researchers collaboratively coded the textual materials, shared their findings, discussed any discrepancies in the coding, and ultimately reached a consensus. Furthermore, the final coding results were discussed by the entire research team to ensure unanimous agreement on the coding outcomes. After the transcription and coding of the data were completed, the results were fed back to some interviewees through interviews, all of whom had no objections to the transcription and coding results.

## 3. Results

### 3.1. Participant Characteristics

Of the 15 interviewees, seven were male, and eight were female, aged 24–67 years, and the general information of the interviewees is shown in Table 2.

### 3.2. Phenomenological Findings

The interviews’ phenomenological analysis revealed four main themes and fourteen related sub-themes. See Table 3.

#### 3.2.1. Undertake Direct and Indirect Multifaceted Economic Pressures

Direct causes:

(1) High drug costs. The costs of chemotherapy, chemotherapy combined with immunotherapy, and the medications used to treat chemotherapy-related adverse reactions are high, making them one of the primary sources of economic burden for patients undergoing chemotherapy.

P11: “The chemotherapy drugs are too expensive each time; the first chemotherapy cost more than 30,000 CNY, and now it’s about 7000 CNY”. P9: “Now the main thing is that this chemotherapy drug is too expensive, two at a time, one is about 6000 CNY, not a penny can be reimbursed, the economic pressure was too heavy”. P1: “Since I got sick, I have used all the chemotherapy, targeted, and immunization drugs, and the drugs I used before were more than 5000 CNY per box, and I need three boxes a month, but the reimbursement was minimal”. P2: “In the past, I had to take an injection to raise white blood cells after each chemotherapy treatment, which cost 1700 CNY each”. P15: “The anti-emetic medicine I buy now costs over 400 CNY for three capsules”.

(2) Other expensive medical expenses. The costs associated with essential hospitalization, examination, deep vein placement, and stoma change during the chemotherapy cycle add to the patient’s financial toxicity to some extent.

P5: “Every month when I come for chemotherapy, the hospitalization fee also costs some money”. P11: “That enhanced CT is quite expensive; I have to do it once every two months, and there are all kinds of routine tests like blood tests”. P12: “I had a PET-CT, which cost me 7800 CNY at a time, and I couldn’t be reimbursed”. P3: “There was a PICC line inserted in my body and a stoma, both of which also cost money every time I changed my medication”.

Indirect causes include the following:

(1) Loss of income and interruption of employment.

P4: “After I got sick, I was completely unable to do my job and completely cut off my financial resources”. P9: “My husband is the only one who works at home, and since I got sick, he has to bring me to the hospital and hasn’t worked for a while”.

(2) High percentage of non-medical costs.

Patients in the chemotherapy stage are hospitalized in cycles, including nutrition, transportation, and accommodation costs accumulated over time, and the costs are immense.

P4: “Coming to the hospital once a month, meals, transportation, and accommodation cost more than 1000 CNY”. P11: “The usual nutrition also needs to be supplemented, which also costs money”.

#### 3.2.2. The Impact of Multidimensional Risk Factors

(1) Health insurance factors.

① Different types and different reimbursement rates. Patients with new rural cooperative medical insurance have low reimbursement rates, and those with employee or commercial insurance will significantly reduce their financial burden.

P5: “I have New Rural Cooperative Medical Insurance, and I can’t report much money each time”. P1: “The current medical insurance is the New Rural Cooperative Medical Insurance with subsistence allowance, so the reimbursement is quite good. If it were only the New Rural Cooperative Medical Insurance, the reimbursement rate would be much lower”. P15: “I have employee health insurance, and the reimbursement is okay; in addition to that, I have commercial insurance. The treatment doesn’t cost anything, and after the health insurance reimbursement, the rest of the commercial insurance is reimbursed”.

② Coverage is limited.

P3: “I feel like we have this kind of big disease; chemotherapy drugs are costly, and we use a lot of drugs that health insurance cannot report. I feel that health insurance does not play a key role”.

③ Low reimbursement rates for out-of-town claims, complicated procedures, and long reimbursement cycles.

P7: “I’m on employee health insurance, and I can get more reimbursement in my hometown, but the reimbursement rate is less over here”. P5: “I’m not local; some expenses need to be reported locally, and I have to advance money back and forth, which is quite a lot, and the reimbursement will only come down in a month or two”. P2: “Some chemotherapy drugs bought at outside pharmacies are reimbursed under the medical insurance, but the procedures are too troublesome”.

④ Insufficient information.

P6: “I don’t know much about the reimbursement process and medical insurance details”.

⑤ Lack of information and trust in commercial insurance.

P9: “I heard someone say that I could buy commercial insurance and get some reimbursement, but I didn’t know anything about it before, so I didn’t think of it”. P4: “Not having bought commercial insurance, I used to resist it, thinking it was a money scam. Now, slowly, there is a process of acceptance”.

(2) Disease-related and family factors.

Patients with pre-cancer underlying diseases and family members who are ill or need to raise children have higher economic pressure, and high family support and economic reserves will reduce their financial toxicity experience.

P1: “In addition to this tumor, there is also diabetes, and I have to take medication every day to control my blood glucose as well”. P1: “My father was diagnosed with lung cancer a year ago, and at that time, the treatment cost over 100,000 CNY a month. He passed away after battling his illness for four years. My mother-in-law’s heart isn’t very good, and she also needs medical care, which costs around 2000 CNY a month”. P8: “I have three children; the oldest is just a freshman in college, and I feel the burden is so heavy (crying)”. P10: “My wife and I encourage each other, and both of us have a particularly good mindset. I am a local resident, my house is being demolished, and I will receive some compensation for the demolition. These savings can cover my medical expenses.

#### 3.2.3. Causing Multiple Adverse Outcomes

(1) Decreased quality of life.

P8: “Now every day feels meaningless; I don’t feel in the mood to do anything”.

(2) Worry about the future and uncertainty.

P7: “I still have some worries about the future. I want to find a job, and my main concern right now is the cost of chemotherapy in the future. “

(3) Depression.

P2: “Looking back now, I feel like I should have gone to Shanghai for the surgery in the first place, like that part of me that metastasized to my liver; the Shanghai side might just mean that it would have been done a little bit cleaner (sighs frequently) and wasted a lot of money”.

#### 3.2.4. Ways of Coping with Financial Toxicity and Expectations

(1) Saving money on daily living expenses.

P13: “It’s costing a lot now; I don’t go out and spend money. My wife dresses and travels more frugally than before”.

(2) Seeking social support and reducing medical costs. This includes turning to relatives and friends for funds and accepting public donations from the community, choosing domestic chemotherapy drugs, buying ostomy bags online, pre-judging medical expenditures, and setting a cap on medical costs.

P1: ”Now I have to borrow money from relatives to see a doctor, and other people in our hometown have also organized a donation for us”. P4: “I can’t afford to use imported medicines so that I can use cheaper domestic ones”. P9: “Now buy that ostomy bag online so that it can be cheaper”. P3: “I set a line for myself; after the cost exceeds this line, I give up and don’t treat it”.

(3) Returning to work to increase my income.

P6: “Although I’m sick now, I still go to work normally, except for that one visit to Shanghai, where I took sick leave. Going to work can earn me some money after all, and the cost of this illness is not small”.

(4) Effective regulation of medical services and donation platforms.

P4: “Some good hospitals also have scalpers and medical scammers, which will also increase the cost of medical care for us patients, so I hope that regulation can be strengthened”. P4: “I don’t believe in platforms like Water Drop Fundraising; I heard it’s a listed company now, and not much money goes to patients. I hope the state can build a good platform, and these donations can be used for needy patients”.

(5) Join a drug clinical trial.

P8: “I hope I can join a drug clinical trial and that drug is free so that I can spend less money”.

## 4. Discussion

### 4.1. Adopt Multidimensional Initiatives to Alleviate Patients’ Economic Toxicity in Response to Specific Causes

(1) Improve pharmaceutical and health policies and accelerate pharmaceutical science and technology innovation. To be specific, improve the evaluation of the benefits of anticancer drugs and enhance the joint decision-making on the costs of doctors and patients. This study found that high drug costs are the most important cause of financial toxicity for patients during chemotherapy, consistent with two other similar studies [14,15]. The reasons may include the following: Chemotherapeutic drugs require multiple treatment cycles. They are more expensive than ordinary drugs. Some imported drugs are even costlier and have low reimbursement rates, leading to high out-of-pocket costs for patients. Moreover, the exorbitant cost of medications far exceeds the income levels of families. Therefore, the government should increase financial support for the pharmaceutical industry, expand the scope of price negotiation for anticancer drugs, and reduce the cost of anticancer drugs, especially imported drugs. On the other hand, the country has increased investment in anticancer drugs and promoted the research, development, and listing of domestic innovative drugs so that patients can use effective and inexpensive anticancer drugs. In addition, to improve the benefit evaluation of anticancer drugs, clinicians can use the 2016 European Society for Medical Oncology Magnitude of Clinical Benefit Scale (ESMO-MCBS) developed by the European Society for Medical Oncology for assessing the size of the clinical benefit of anticancer drugs so that limited medical care can be provided. Concerning the benefit magnitude to rationalize allocating limited medical and personal resources [16], the ESMO-MCBS is used to assess the magnitude of the clinical benefit of anticancer drugs, enabling the rational allocation of limited medical and personal resources. Increasing cost communication between doctors and patients, and choosing the most appropriate treatment plan in the context of the actual situation can reduce the cost of treatment for patients [17].

(2) Enhance the medical staff’s professional competence and improve the quality evaluation system. This study found that other medical-related costs aggravated patients’ financial toxicity to a certain extent. The reasons for this may be that patients need multiple cycles of chemotherapy, repeated hospitalization, some routine blood or CT tests before chemotherapy, nausea and vomiting, and other adverse reactions after chemotherapy. In addition, some patients will be retained with a deep-vein catheter and stoma, which need to be cared for regularly, and all of these procedures will aggravate the financial toxicity of the patients. Medical personnel should improve their business ability. They need to strengthen the monitoring and management of patients’ conditions. It is important to actively promote continuity of care services. They should reduce the occurrence of adverse reactions and complications. Medical staff must use clinical pathways rationally. They need to regulate diagnostic and therapeutic behaviors strictly. Additionally, they should reduce unnecessary medical checkups during the diagnostic and therapeutic process.

(3) Improve the paid vacation and employment protection systems and promote the development of telemedicine services. Indirect causes of unemployment or employment disruption among patients and their families are particularly prominent. The reason for this analysis may be that chemotherapy is cyclical and accompanied by a series of toxic side effects, which can lead to limitations in the patient’s ability to work. In contrast, the patient’s family members need to take leave or quit their jobs to take care of the patient frequently, the family loses its source of income, and the patient’s financial toxicity is aggravated. This is consistent with Yuan Fang et al.’s [18] study that employment status may be economic stress’s most important socioeconomic indicator. Therefore, the government should improve the paid leave and employment protection systems for patients to reduce the economic loss caused by unemployment due to illness. In future interventions for colorectal cancer populations during chemotherapy, it is important to understand the difficulties they face in returning to work and to assist in returning to work, as well as making appropriate job adjustments to reduce their financial toxicity. The patient is the subject of past research on financial toxicity. Still, there are fewer studies in China on the impact of the employment status of family caregivers of cancer patients on their financial toxicity, and the path of the role relationship can be further explored in the future. In addition, the community can organize lectures or information sessions to teach patients financial management skills, budgeting, and ways to seek assistance. The proportion of non-medical costs of colorectal cancer patients in the chemotherapy stage should not be ignored. Telemedicine services can also provide patients with medical information and services and reduce patients’ travel expenses, time, and other losses to control patients’ indirect costs. In the future, the government can promote the breakthrough of new technologies to promote the popularization, process, and refinement of telemedicine in China [9].

### 4.2. Effectively Respond to and Control the Influencing Factors to Reduce the Financial Toxicity of Patients

(1) Expand the scope of medical insurance reimbursement, increase the reimbursement rate, reduce regional disparities in healthcare and establish a diversified medical insurance system. This study found that different types of medical insurance affect the economic burden of patients’ treatment. Rural patients have limited healthcare resources and low levels of medical care. When choosing to seek treatment at large hospitals, the reimbursement ratio for various insured medications under the New Rural Cooperative Medical Insurance is low, leading to a higher financial toxicity for these patients. This finding is consistent with the research by Ma Xiaoqi et al. [19]. Furthermore, patients’ financial toxicity is also influenced by the dissemination of insurance knowledge, policies, and systems, which aligns with the findings of Sun Yanling et al. [20]. The severity of financial toxicity may be related to limited insurance coverage, low reimbursement ratios for out-of-area treatments, complicated procedures, long reimbursement cycles, and a lack of insurance information. Therefore, to reduce regional disparities in healthcare, the government should strengthen the construction of medical facilities in remote rural areas, develop and implement public health policies tailored to different regions, and regularly assess the distribution and quality of medical services to ensure that every region receives adequate medical resources and covers all populations. Meanwhile, the government can improve the reimbursement ratio of medical insurance, accelerate the efficiency of medical insurance reimbursement, and expand the publicity of medical insurance policy information. In addition to basic medical insurance, the government should improve primary medical insurance and outpatient chronic disease reimbursement, optimize the starting line of medical insurance reimbursement, and adjust the allocation of medical resources [21]. In addition, this study shows that the cost impact of commercial insurance for cancer patients is less, but most patients have insufficient trust and awareness of commercial insurance. Commercial insurance is not well recognized by society in China. The government should introduce incentives to develop commercial insurance. Commercial insurance companies should utilize their advantages in actuarial services and claims handling. They need to develop more mature cancer insurance to alleviate patients’ economic burden [22]. Commercial insurance organizations should share their information with the public. This way, those in need can choose appropriate insurance. The government should strengthen its efforts to explore establishing a diversified medical insurance system, give full play to the complementary role of charitable medical assistance for cancer patients, develop charitable medical assistance practices, assist charitable medical assistance programs to carry out, and strengthen the supervision of charitable medical programs [23]. At the same time, it should strengthen the supervision of charitable medical programs and reduce the economic burden of patients from multiple angles.

(2) Improve cost health literacy, carry out early screening for financial toxicity, and enhance family support systems. The economic pressure is heavier for patients who have underlying diseases before cancer, have sick family members, or need to raise children. The reasons analyzed were that having an underlying disease before cancer and routine treatment put the patients under some economic pressure, in addition to the fact that middle-aged patients were predominant in this study. They usually took the responsibility of caring for their elderly parents, raising children, and paying for daily living expenses, which led to a further increase in financial toxicity. This differs from the research conducted by Wang Lingling et al. [24], who stated that the financial toxicity for the elderly is higher than that of middle-aged and young individuals. This may be related to the lack of savings or income sources among the elderly, the financial burden of long-term treatment for chronic diseases, and the non-reimbursable costs of long-term care. Therefore, future research may need to conduct longitudinal or intervention studies in order to understand the complex topic of financial toxicity. At the same time, it should inspire healthcare professionals to not only consider age as a variable when addressing population heterogeneity but also include factors such as social background, educational level, and disease experience. Healthcare professionals can help reduce financial toxicity by improving patients’ health literacy and price-related competency levels and promoting patients’ self-efficacy about the costs of managing underlying diseases and cancer. In addition, in focusing on the overall economic status of the patient’s family, financial toxicity data can be collected for early screening and dynamic monitoring of cancer financial toxicity [25]. Patients with high family support and financial reserves mitigate their experience of financial toxicity, which is consistent with previous studies [5,26]. Consistent with past research, healthcare professionals should enhance patients’ family support, and good family support can reduce patients’ sense of shame and promote positive perception and coping with the disease [27].

### 4.3. Emphasize Psychological and Emotional Health and Improve Positive Psychological Experiences

This study found that colorectal cancer patients in the chemotherapy stage have a significant psychological burden, which is prone to making them experience anxiety, depression, and other adverse emotions. The reasons may be related to the increase in medical expenses due to the treatment of the disease, the impact on the work of individuals and family members, the increase in the family’s financial burden, and the patient’s uncertainty about the prognosis. According to the theoretical model of economic burden after cancer diagnosis [11], psychoeconomic burden is further categorized into concern about future costs and reflection on past and current financial burdens. The model differs from previous studies by distinguishing the subtypes of psychoeconomic burden more precisely, and targeted interventions can be implemented for specific subtypes to achieve better intervention effects. Worry and anxiety triggered by patients’ future costs were predominant in this study. Therefore, this suggests that more attention should be paid to patients’ psychological conditions during clinical practice, and group communication activities and psychological counseling services should be provided to alleviate their subjective feelings about financial toxicity. Encouraging patients to share their experiences and feelings about their struggle with cancer and strengthening communication with the outside world can not only alleviate their own adverse emotions but also provide a reference for patients’ economic decision making [28]. Some studies [29] suggest that providing price transparency and financial navigation in an appropriate setting can also reduce patients’ anxiety about cancer treatment.

### 4.4. Strengthen the Supervision of Corresponding Service Platforms to Protect Patients’ Legitimate Rights and Interests

This study found that medical scammers and scalpers seize high-quality medical resources, seek undue wealth, and aggravate patients’ medical care costs. Donation platforms lack transparency and cannot be trusted. Therefore, stricter laws and regulations should be formulated to explicitly prohibit scalping behavior, increase penalties, and use Artificial Intelligence to monitor and identify scalping activities for timely enforcement. The government can establish an open and transparent information system so that the public can inquire about the use of fundraising and the flow of funds, publish regular reports on the use of donations, enhance public trust in donation platforms, and introduce blockchain technology to make donation information tamper-proof and track the flow of funds.

## 5. Conclusions

This study identifies the primary sources of financial toxicity in colorectal cancer patients undergoing chemotherapy, various influences, adverse outcomes, and patient coping styles and expectations, providing a basis for developing effective ways of dealing with financial toxicity and for the future identification of patients with high risk for financial toxicity. This study found that colorectal cancer patients undergoing chemotherapy have expectations regarding alleviating their financial toxicity, which is not mentioned in the theoretical model. Future research could expand and enrich the theoretical model and validate it in clinical practice. This study has some limitations: the sample was from one tertiary care hospital, information bias may exist, and this study failed to break down the experience of different chemotherapy cycle groups; participants’ responses to financial toxicity may have been influenced by recall bias. The sample size was small, and the distinction for subtypes of psychoeconomic burden was unclear. These limiting factors may have led to the insufficient extraction of research findings on the economic toxicity theme for patients. In the future, multicenter, longitudinal studies and mixed-methods research could be conducted to delve deeper into the factors influencing financial toxicity in colorectal cancer patients undergoing chemotherapy during different treatment periods, identify targets for intervention, and provide personalized interventions. This study was conducted only from the patient’s perspective. Future studies can be performed from the perspectives of healthcare professionals, family caregivers, and other personnel. A standardized management model and a unified clinical protocol can be established to manage the economic risks of chemotherapy-stage colorectal cancer patients throughout the entire process and reduce economic risks’ impact.

## Figures and Tables

**Table 1 curroncol-32-00023-t001:** Interview outline.

Main Topics	Questions
FT experience on oneself	(1) Could you please briefly describe your experience of diagnosis and treatment?
	(2) What was your financial situation before chemotherapy for colorectal cancer? Has there been any change in these areas after receiving chemotherapy? Please expand on this.
	(3) How have these changes affected your life?
	(4) How did you deal with or cope with these financial changes?
	(5) Is there anything else you would like to add?

**Table 2 curroncol-32-00023-t002:** Characteristics of participants (*N* = 15).

Number	Gender	Age (Years)	Educational Attainment	Current Address	Lesion Location	Disease Staging	Number of Chemotherapy Sessions	Monthly per Capita Household Income (CNY)	Type of Medical Insurance
P1	Female	45	illiteracy	countryside	colon	IV	43	②	①
P2	Male	29	vocational secondary school	city	colon	IV	24	②	②
P3	Male	60	undergraduate	city	colon	IV	8	④	③
P4	Male	41	junior high school	countryside	colon	IV	6	①	①
P5	Female	24	junior high school	countryside	colon	II	12	③	①
P6	Female	30	undergraduate	city	rectum	IV	31	④	③
P7	Female	51	junior college	city	colon	II	4	④	③④
P8	Female	42	junior high school	countryside	colon	IV	16	①	①
P9	Female	53	illiteracy	countryside	rectum	IV	13	③	①
P10	Male	41	undergraduate	city	colon	IV	11	④	③
P11	Female	41	junior high school	countryside	colon	IV	5	①	①
P12	Male	67	junior school	countryside	colon	IV	6	④	③
P13	Male	55	junior high school	countryside	rectum	IV	35	④	③④
P14	Male	39	undergraduate	city	rectum	IV	3	④	③
P15	Female	52	senior middle school	city	colon	III	4	④	③④

Note: Per capita monthly household income (CNY): ①: <2000; ②: 2000~; ③: 3000~; ④: 4000~; type of medical insurance: ① New Rural Cooperative Medical Insurance; ② Urban Residents’ Medical Insurance; ③ Urban Workers’ Medical Insurance; ④ Commercial Medical Insurance.

**Table 3 curroncol-32-00023-t003:** Themes and sub-themes of economic toxicity experience in patients undergoing chemotherapy phase of colorectal cancer.

Themes	Sub-Themes
Undertake direct and indirect multifaceted economic pressures	Direct causes
	Indirect causes
The impact of multidimensional risk factors	Health insurance factors
	Disease-related and family factors
Causing multiple adverse outcomes	Decreased quality of life
	Worry about the future and uncertainty
	Depression
Ways of coping with economic toxicity and expectations	Saving money on daily living expenses
	Seeking social support and reducing medical costs
	Returning to work to increase my income
	Effective regulation of medical services and donation platforms
	Join a drug clinical trial

## Data Availability

The original contributions presented in the study are included in the article.

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
