# Peer review of "The Financial Toxicity Experience of Patients with Colorectal Cancer During Chemotherapy: A Qualitative Study"

_curroncol, 2024, doi:10.3390/curroncol32010023_

Round 1

Reviewer 1 Report

Comments and Suggestions for Authors

This is a qualitative study exploring the economic toxicity for patients with colorectal cancer during chemotherapy. Five themes were identified through interviews with 15 participants identified through purposeful sampling. The results are presented with quotes to support the themes. The discussion takes a different approach identifying multidimensional initiatives to alleviate patients’ economic toxicity.

The abstract has good descriptions of the purpose of the study, methods,  analysis , and results. The conclusions are appropriate though the last sentence does not flow from the actual interviews as presented (In the future, healthcare professionals should identify patients at high risk for economic toxicity, provide economic toxicity interventions and support, and mitigate their exposure to economic toxicity). These appear to be the authors’ recommendations based on the findings rather than a conclusion of the actual findings.

The introduction provides a good background on the prevalence of colorectal cancer and the economic consequences. They note that while there have been studies in other countries, there has not been many studies in China. A theoretical framework was chosen for the study and is briefly described which can be used in qualitative studies. I do not have recommendations for change.

The methods include the descriptions of the participants and recruitment efforts. The inclusion and exclusion criteria are clearly described and appropriate to the study. The rational for the sample size is well stated with saturation of data occurring with the 13th patient and two additional patients interviewed. It is not clear why there were two additional patients. It may be the two patients who were pre-interviewed, but it is not clear.

The design is described in good detail. The questions used to structure the interview were provided, which is helpful. The process of the interviews was described in detail. The transcription and analysis processes are appropriate to the methods. The coding process is clearly noted. There was confirmation by the study group which increases validity of the findings.

The results are presented well with a description of the participants and the themes and subthemes. There is good evidence of the coding presented in the quotes from the participants. Direct costs were identified and coded into subthemes. The multidimensional risk factors appear to be health insurance factors and disease-related and family factors. The numbering and bolding need some attention on page 5. The adverse outcome subthemes are clear as are the ways of coping.

The discussion focuses on initiatives to alleviate patients’ economic toxicity, but this does not appear to be identified by the patients in the interviews, but by the authors. While these are important initiatives and may be very helpful, this is not a discussion of the findings and lacks the comparison to the other literature. It is understandable that these are changes that are needed, however they don’t appear to be identified directly by the participants. Some discussion of the results should be included before the discussion of what is needed to alleviate economic toxicity.

Limitations are stated well and appear complete.

Reviewer 2 Report

Comments and Suggestions for Authors

General Comments:

The manuscript investigates an important topic, economic toxicity in colorectal cancer patients undergoing chemotherapy. The study's descriptive qualitative approach is appropriate and the findings provide valuable insights into patient experiences. The manuscript is well-structured and supported by thematic analysis, although certain areas require improvement for greater clarity and robustness.

Strengths:

The study addresses a critical yet underexplored aspect of oncology care, especially in China, where relevant research is limited.

The use of thematic analysis is appropriate for qualitative data.

The inclusion and exclusion criteria are well-defined, ensuring relevance and data quality.

Clear division of themes and subthemes enhances readability.

Direct quotes from participants strengthen the authenticity of the findings.

The recommendations for policy and practice are well-articulated, linking findings to actionable interventions.

Weaknesses:

The sample size of 15 participants, while achieving data saturation, limits the generalizability of the findings. Expanding the study to multiple centers could enhance robustness.

The sample lacks diversity in socioeconomic and geographic contexts, which could influence experiences of economic toxicity.

The paper does not describe the process of validating the thematic analysis, such as the use of inter-coder reliability measures.

While participant demographics are presented, their influence on specific findings (e.g., income level, insurance type) is underexplored.

The discussion could better integrate the results with existing literature to highlight similarities and deviations.

The limitations section acknowledges some constraints but does not provide enough emphasis on how these might influence the study’s conclusions.

Some sentences are long and complex, making them harder to follow. For example, in the discussion, several paragraphs could be restructured for conciseness.

Recommendations:

Clearly outline the steps taken to ensure the rigor of the thematic analysis, such as double-coding or peer debriefing.

Discuss how potential biases, like researcher influence, were mitigated during interviews and analysis.

Provide a more detailed examination of how participant demographics (e.g., age, income, insurance type) influence the identified themes.

Explore regional healthcare disparities that may affect the findings.

Include a comparative analysis with similar studies to contextualize findings within broader literature.

Suggest avenues for future research, such as longitudinal studies or intervention trials.

Revise lengthy sentences for clarity.

Address typographical errors and ensure consistency in style.

Adding visual aids, such as thematic maps or participant quotes categorized by themes, could enhance reader engagement and comprehension.

Reviewer 3 Report

Comments and Suggestions for Authors

General comments:

The topic of this study – the economic side-effects or “economic toxicity” of chemotherapy – is very important, and thus, the study would be of interest to medical professionals who care for cancer patients. The paper would be especially relevant to hospital social workers and psychological counselors who are involved in the patient-care process.

My main critical comment is that, in making recommendations (in section 4), the authors seem to implicitly assume that their findings are from a large, representative sample, when in fact, the sample is, at best, a small “purposive” sample (line 65). To be sure, the authors later acknowledge the limitations of their approach (lines 391-394). However, this acknowledgment of limitations comes rather late in the paper. In my opinion, the authors should, in the paper’s introduction (around line 52, when the “theoretical model” is first mentioned), state up-front that they are conducting an exploratory investigation that might provide a foundation for the kinds of research they describe on lines 394-400.

My other critical comment is that, in the paper’s concluding section, the authors might make some suggestions, based on their results, about how the “theoretical model” (lines 52-53) might be improved over the one advanced by Zafar and Abernethy. The authors might also discuss how the model of Zafar and Abernethy, which is based on the US medical system, might be modified in order to be applicable to the PRC medical system or the medical system(s) of other societies.

Specific comments:

Line 20 – clarify “ways and expectations”

Line 30 – “rank third”: what are 1 and 2?

Line 39 – define “economic toxicity” before introducing the term

Lines 42-43 – “Zafar et al.” should be “Zafar and Abernethy” [see reference 7]

Line 47 – say “International” instead of “Foreign”

Line 48 – capitalize “The” at the beginning of the sentence

Line 48 – say “personal” instead of “inner”

Line 50 – capitalize “China”

Lines 52-59 – the sentence is too long; use numbered points to guide the reader

Lines 42-62 – do not use “financial” and “economic” interchangeably; the term, “economic toxicity” seems more general than “financial toxicity,” so the authors should clarify their choice of terminology (“financial” or “economic”? pick one and be consistent throughout the paper)

Line 77 – begin the sentence with “We adopted …”

Line 78 – capitalize “We” at the beginning of the sentence

Lines 89-90 – be more specific about where the interviews took place, briefly describing the range of locations

Lines 91-95 – in this sentence, say, “… the interviewer was instructed to repeat, reorganize, …”

Line 115 – the table title should refer to “background characteristics”; “demographic” is a term used in reference to a population

Lines 120ff – in the findings section (3.2), the discussion of patient expenditures would be enhanced if the amounts were also expressed as a percentage of the patient’s monthly or yearly household income

Line 161 – reconsider using boldfaced type

Line 206 – in section 3.2.3., make sure that “P” for patient number is capitalized

Lines 243ff – this section (4) should be retitled “Recommendations” instead of “Discussion”

Line 247 – begin the sentence, “Specifically, improve the evaluation…”

Line 250 – say “This is consistent with two other similar studies …”

Lines 263-264 – a sentence fragment, beginning with “benefit magnitude” needs correction

Line 275 – say “... all of these procedures will aggravate …”

Lines 303-304 – say “… control patients’ indirect costs …”

Line 313 – remove “Consistent”

Line 314 – say “coverage” [lower-case “c”]

Line 316 – say “ameliorate” rather than “affect”

Line 328 – a period (not a comma) should follow [21]

Line 333 – clarify “to land”

Line 350 – say “past research”

Line 356 – “more significant burden …”: more significant than … what?

Line 360 – provide a supporting reference for “the theoretical model of economic burden”

Line 381 – capitalize “Artificial Intelligence”

Line 382 – clarify who should “Establish an open and transparent information system …”

Lines 383-384 – clarify who should “publish … enhance … and introduce …” Are the authors referring to the public sector? private sector? both?

Line 391 – a colon [:] not a semicolon [;] should follow “limitations”

Line 391 – say “one hospital” [not “1 hospital”]

In closing, I think the authors have conducted a worthwhile study. I believe the paper would be strengthened if the authors addressed the above general comments and specific (mostly editorial) comments.

Comments on the Quality of English Language

Routine copy-editing is needed

Round 2

Reviewer 2 Report

Comments and Suggestions for Authors

The paper has been improved significantly, and I suggest to be accepted in the present form 

Reviewer 3 Report

Comments and Suggestions for Authors

The authors have worked hard to address my concerns, and the efforts have resulted in a paper that is noticeably improved over the previous version.

Comments on the Quality of English Language

OK -- but copyediting is still needed